# MG-*Pe*: A Novel Galectin-3 Ligand with Antimelanoma Properties and Adjuvant Effects to Dacarbazine

**DOI:** 10.3390/ijms23147635

**Published:** 2022-07-11

**Authors:** Stellee M. P. Biscaia, Cassiano Pires, Francislaine A. R. Lívero, Daniel L. Bellan, Israel Bini, Silvina O. Bustos, Renata O. Vasconcelos, Alexandra Acco, Marcello Iacomini, Elaine R. Carbonero, Martin K. Amstalden, Fábio R. Kubata, Richard D. Cummings, Marcelo Dias-Baruffi, Fernanda F. Simas, Carolina C. Oliveira, Rilton A. Freitas, Célia Regina Cavichiolo Franco, Roger Chammas, Edvaldo S. Trindade

**Affiliations:** 1Department of Cellular Biology, Federal University of Paraná (UFPR), Curitiba 81531-980, Brazil; stellee.biscaia@gmail.com (S.M.P.B.); daniellimabellan@gmail.com (D.L.B.); israelbini@gmail.com (I.B.); fernanda.simas@gmail.com (F.F.S.); krokoli@ufpr.br (C.C.O.); crcfranc@ufpr.br (C.R.C.F.); 2Department of Chemistry, Biopol, Federal University of Paraná (UFPR), Curitiba 81531-980, Brazil; cassiano.ufpr@gmail.com (C.P.); rilton@ufpr.br (R.A.F.); 3Post-Graduate Program in Medicinal Plants and Phytotherapics in Basic Attention, Parana University (UNIPAR), Umuarama 87502-210, Brazil; francislaine@prof.unipar.br; 4Department of Radiology and Oncology, Faculty of Medicine, Center for Translational Research in Oncology (CTO), Cancer Institute of the State of São Paulo, University of São Paulo (USP), São Paulo 01246-000, Brazil; silvinabvg@gmail.com (S.O.B.); reottes@gmail.com (R.O.V.); 5Department of Pharmacology, Federal University of Paraná (UFPR), Curitiba 81531-980, Brazil; aleacco@ufpr.br; 6Department of Biochemistry and Molecular Biology, Federal University of Paraná (UFPR), Curitiba 81531-980, Brazil; iacomini@ufpr.br; 7Institute of Chemistry, Federal University of Catalão (UFCAT), Catalão 75704-020, Brazil; elainecarbonero@ufcat.edu.br; 8Department of Clinical Analyses, Toxicology and Food Sciences, School of Pharmaceutical Sciences of Ribeirão Preto, University of São Paulo (USP), Ribeirão Preto 14040-903, Brazil; martin.amstalden@gmail.com (M.K.A.); fabiokubata@gmail.com (F.R.K.); mdbaruff@fcfrp.usp.br (M.D.-B.); 9Beth Israel Deaconess Medical Center, Harvard Medical School, Boston, MA 02115, USA; rcummin1@bidmc.harvard.edu

**Keywords:** galectin-3 Lectin, melanoma, oxidative stress, dacarbazine, quartz crystal microbalance (QCM)

## Abstract

Melanoma is a highly metastatic and rapidly progressing cancer, a leading cause of mortality among skin cancers. The melanoma microenvironment, formed from the activity of malignant cells on the extracellular matrix and the recruitment of immune cells, plays an active role in the development of drug resistance and tumor recurrence, which are clinical challenges in cancer treatment. These tumoral metabolic processes are affected by proteins, including Galectin-3 (Gal-3), which is extensively involved in cancer development. Previously, we characterized a partially methylated mannogalactan (MG-*Pe*) with antimelanoma activities. In vivo models of melanoma were used to observe MG-*Pe* effects in survival, spontaneous, and experimental metastases and in tissue oxidative stress. Analytical assays for the molecular interaction of MG-*Pe* and Gal-3 were performed using a quartz crystal microbalance, atomic force microscopy, and contact angle tensiometer. MG-*Pe* exhibits an additive effect when administered together with the chemotherapeutic agent dacarbazine, leading to increased survival of treated mice, metastases reduction, and the modulation of oxidative stress. MG-*Pe* binds to galectin-3. Furthermore, MG-*Pe* antitumor effects were substantially reduced in Gal-3/KO mice. Our results showed that the novel Gal-3 ligand, MG-*Pe,* has both antitumor and antimetastatic effects, alone or in combination with chemotherapy.

## 1. Introduction

One of every six deaths worldwide is due to cancer, which took the lives of 10 million people in 2020 [1]. Melanoma is the least prevalent [2] type of skin cancer, yet it presents the highest mortality rate due to its metastatic capacity [3].

Current treatments for melanoma include chemotherapy, immunotherapy, topical, targeted therapy, and nanotechnology [4]. The first approved chemotherapy for melanoma was Dacarbazine (DTIC) [5], which is still used in low and middle-income countries, such as Brazil [6,7,8], despite the several adverse effects [9]. Cancer sensitivity to therapeutic agents is conditioned to several factors, such as tumor microenvironment (TME) and drug resistance, by means of intratumor cell plasticity [10,11,12]. The intratumor heterogeneity allows some tumor cells to survive the treatment, developing a therapy-resistant cell phenotype, leading to relapse [13].

Nonetheless, in a meta-analysis, Jardim and colleagues demonstrated that the combination of non-toxic drugs compared to monotherapy prolongs the survival, resulting in synergism between these molecules [14]. Thus, finding non-cytotoxic drugs that could increase the survival of metastatic melanoma patients would be an asset.

Our group has described non-toxic compounds with antitumor properties, a partially methylated mannogalactan (MG-*Pe*), extracted from the edible mushroom *Pleurotus eryngii*. MG-*Pe* showed a significant antimelanoma activity modulating in vitro (B16-F10 murine cell) malignancy-related cell processes, such as cell invasion, and reducing solid tumor development in vivo without toxicity [15].

One of the pathways of malignant transformation and tumor progression is associated with changes in the expression of molecules, such as extracellular galectin-3 (Gal-3), and its binding to other molecules, leading to a process of tumor growth stimulation [16]. Gal-3 is a β-galactoside-binding lectin, overexpressed in melanoma, and is a marker of progression in melanocytic lesions and prognosis in primary melanoma patients [17]. Extracellular Gal-3 contributes to metastasis and melanoma progression [18,19,20] and is involved in multidrug resistance [21]. Furthermore, Gal-3 protects tumor cells from cell death in solid tumors under hypoxia and nutrient deprivation. Some studies indicate that Gal-3 is associated with several cell invasion and metastasis stages, such as angiogenesis, cell-extracellular matrix interaction, dissemination by blood flow, and extravasation [22,23,24]. This overexpression of extracellular Gal-3, under stressful conditions, appears to be part of an adaptive process that leads to tumor cell survival [25]. Gal-3 is also very important in modulating the immune system in neurodegenerative diseases, by improving inflammatory condition, modulating brain innate immune responses in the brain, and on microglial activation patterns in physiological and pathophysiological settings in a context-dependent manner in microglia polarization [26,27,28].

Thus, the inhibitors of Gal-3 have demonstrated effects in reducing lung metastasis, proliferation, and tumor growth and exhibit synergic effects with chemotherapy [29,30]; clinical studies are ongoing with melanoma patients, using Gal-3 inhibitor GR-MD-02 and ipilimumab (Clinical trial NCT02117362) [31]. Gal-3 inhibitors demonstrated effects in reduction in lung metastasis, proliferation, and tumor growth, and their synergic effect with chemotherapy [29] this cell survival was also related to oxidative stress [32].

When overly produced reactive oxygen species (ROS) causes oxidative stress, which is involved in the promotion of tumorigenesis and tumoral progression [33,34,35], induced genetic and epigenetic alterations, DNA damage, and lipid peroxidation (LPO) [36]. On the other hand, in normal conditions, ROS are important for the regulation of physiological cellular functions [37]. Whereas high levels of ROS can cause cellular oxidative stress and induce apoptosis, low levels of ROS can promote the cell cycle transition in various cellular systems, stimulating cell proliferation [36]. With the oxidative stress state, antioxidants, such as reduced glutathione (GSH), are increased in melanoma more than what is observed in the other skin tumors. Melanoma cells are therefore well-adapted to oxidative stress [38].

Here, we characterized MG-*Pe* as a novel extracellular Gal-3-binding polysaccharide and described its anti-melanoma effects of MG-*Pe*. We also showed that these effects were likely dependent on the interaction of MG-*Pe* and Gal-3 since the polysaccharide was non-effective in Gal-3 KO animals.

## 2. Results and Discussion

### 2.1. Adjuvant Effect and Increased Survival of Tumor-Bearing Mice Treated with MG-Pe

We previously demonstrated that the administration of partially methylated mannogalactan extracted from mushroom *Pleurotus eryngii* (MG-*Pe*) (50 mg/kg) inhibited primary melanoma tumor growth (*versus* control PBS) without inducing toxicity [15]. Here, in this study, we investigated MG-*Pe* treatment versus dacarbazine (DTIC—chemotherapy agent with known antitumor effect), alone or in combination (adjuvant effect). As expected, our results reconfirmed (Figure 1A) that MG-*Pe* alone has an antitumoral effect. Likewise, the treatment of tumor-bearing mice with DTIC alone also reduced melanoma tumor growth by 66%, as compared to PBS. However, we did not observe any statistical difference between treatment with DTIC alone to MG-*Pe* alone (Figure 1A). This was an unexpected result, as such a non-toxic treatment generated a similar antitumor effect compared to a chemotherapy agent.

More importantly, the DTIC+MG-*Pe* combined group showed impaired tumor growth starting at day 11 compared to the control (PBS). Moreover, at the end of the experiment, DTIC+MG-*Pe*-treated animals developed the smallest tumors (Figure 1A), decreasing 83% compared with PBS and 51% compared with DTIC.

These results suggest an opportunity that may directly benefit patient survival and quality of life. Thus, we also evaluate survival outcomes with the same groups. The DTIC alone increased survival by 46%, but the combined treatment showed an even better effect, increasing the survival of animals by 54% (median survival) versus PBS (Figure 1B).

These results were promising and indicated that MG-*Pe* could serve as an alternative for therapy, such as the galectin inhibitor galactomannan, which sensitized solid tumors and colon cancer cells to chemotherapy, and it is currently under clinical study [39,40]. Moreover, DTIC is still recommended to be used as a first line therapy agent in low and middle-income countries for melanoma treatment, where no targeted therapies or immunotherapy regimens can be prescribed due to their prohibitive costs [6,7,8,41,42]. Moreover, it corroborates with the literature showing that polysaccharides may have synergistic antitumor action with chemotherapeutic agents [43]. Thus, finding an agent with an adjuvant effect to DTIC that does not cause more adverse effects and still potentiates the antitumor effect may have a positive social impact on the management of melanoma patients.

### 2.2. MG-Pe Reduces Experimental and Spontaneous Metastasis Development

Tumor metastasis is the leading cause of death in patients with metastatic melanoma [3]. Thus, we evaluated the effects of MG-*Pe* on experimental and spontaneous metastasis in animals.

In the experimental lung metastasis model, MG-*Pe* (alone) and DTIC (alone) reduced metastatic *foci* in the lung (43% and 46% respectively) when compared to PBS (Figure 2A,A1), showing the antitumoral effect of MG-*Pe* and corroborating with the data in the subcutaneous tumor growth model (Figure 1A). Additionally, when administered together as MG-*Pe* + DTIC, it reduced metastatic *foci* in the lung (53%), showing an adjuvant effect (Figure 2A and lung images in Figure 2A1), corroborating with the data in Figure 1A, which also shows MG-*Pe* an antitumor action. For the spontaneous metastasis model (Figure 2B), tumors (solid subcutaneous) were allowed to develop; the animals were then treated for ten days (as described in Section 3.3.1). Tumors were then surgically removed, and after one month (without treatment) the metastases were evaluated in both groups (recurrence was analyzed in different organs and tissues). In this qualitative experiment we found that only one treated animal had a recurrence (13%) at the same site as the removed primary tumor, compared to untreated animals that showed multiple metastatic *foci* at different sites as reported below and visualized in Figure 2B.

We evaluated the recurrence of primary tumors in all experimental groups. In the PBS group, 67% of mice (4 of 6 mice) showed tumor development at the surgical removal site (around the 14th day after surgery). By contrast, in the MG-*Pe* treated group, only 13% of the mice (1 of 8) presented recurrence, on the 16^th^ day post-surgery. Moreover, for the control group (PBS), the evaluation of all other organs indicated diverse metastatic *foci* (Figure 2B) presented metastatic *foci*: in the lungs of 33% of mice (three of six); in the heart (17% of mice—one of six); axillary lymph node chain (17% of mice—one of six); and in the same animal there were secondary tumors (17%) in the lung, heart and lymph nodes. Moreover, at the end of the experiment, splenic hemorrhage was observed in the PBS group (50%, three out of the six mice in the PBS group). In contrast, as mentioned above, the MG-*Pe* treated group showed only one of eight mice with recurrence of the primary tumor (13%) and three of eight with a hemorrhagic spleen (38%). Strikingly, no macroscopic metastatic lesions were observed in all evaluated organs of the group treated with MG-*Pe*. These data indicate that MG-*Pe* has a more effective and prolonged antimetastatic action, even after one month from the end of the treatment period. Treatments that reduce the metastatic potential of tumor cells are of great interest and one of the main objectives of current oncological studies [44]. Other polysaccharides with antitumor activity have been described in the literature. Examples include (i) a sulfated homogalactan, which was able to reduce the growth of solid tumors and metastasis, without inducing side effects in mice [45] (ii) a sulfated heterorhamnan, which decreased metastatic capacities, such as migration and invasion of B16-F10 cells [46]; (iii) a 3-O-methylated heterogalactan, which modulated immunological functions [47]. All these processes interfere with cellular interactions within the tumor microenvironment (TME) and modify the dynamics of the metastatic process. Among these important changes that occur in the tumor environment (TME) is the overexpression and secretion of Gal-3, which is associated with several stages of cell invasion and metastasis, leading to angiogenesis, cell-extracellular matrix interaction, intravasation, vascular dissemination, and extravasation [22].

### 2.3. MG-Pe Binds Gal-3

Given that galectins bind glycans contain galactose moieties, and MG-*Pe* has galactose as a component [15], we sought to assess whether Gal-3/MG-*Pe* interaction could mediate the antitumor effects of this polysaccharide. We, therefore, evaluated MG-*Pe* direct binding to Gal-3 using physicochemical methods. To test this hypothesis, different quantitative and qualitative methodologies were used.

Because of the large size and polydispersity of MG-*Pe*, several approaches were used to evaluate binding to Gal-3. As seen in Figure 3A,B, we measured the contact angle between water and surface of the gold sensor, under different conditions, to confirm each functionalization step due to surface-free energy modification, from the sensor alone, the chemical modifications with *β*-mercaptoethanol, EDC/NHS, and Gal-3. The contact angle modification, due to the adsorption binding of MG-*Pe* and lactose can be observed in the last images. The results shown in Figure 3A indicate the differences between each step, by the contact angle (left column) and also by the roughness (AFM) (right column), showing that each step of the chemical functionalization was confirmed. Moreover, in Figure 3B, the results show a contact angle to the left and the roughness to the right, which indicates that Gal-3 interacts to both lactose and MG-*Pe*.

When we compare the samples with lactose, a known Gal-3 ligand [48], concerning our target (MG-*Pe*), the roughness increased 65% in the lactose group compared to a 170% increase in the MG-*Pe* group. These results indicate that lactose and MG-*Pe* bind to Gal-3. This result was expected, given the difference in the mass of the two molecules, MG-*Pe* is a polysaccharide with the mass of 20,900 g/mol, and lactose has a mass of 342.3 g/mol.

Moreover, QCM analysis (Figure 3C) corroborates these other results and shows that MG-*Pe* binds to Gal-3 in all tested concentrations. When MG-*Pe* was tested in the highest concentration, 81% more mass was adsorbed in the sensor than lactose. Altogether, these results demonstrated that MG-*Pe* binds to Gal-3 ligand. To summarize these results, Figure 3D shows a schematic drawing represents the binding of polysaccharides and lactose with Gal-3 immobilized on the gold sensor after chemical modifications and differences between the molecules.

Gal-3 has been described as being overexpressed widely in tumor cells and its expression is correlated with cancer aggressiveness and metastasis [22]. Gal-3 is one member of the broad family of galectins, all of which share the ability to bind simple β-galactosides and have a significant sequence similarly in their overall carbohydrate-binding domain (CBD) [48]. However, galectins may also bind to other saccharides through distinct sites, which has a potentially significant biological consequence. For example, Galectin-1 (Gal-1) binds to Davanat, an α-galactomannan, at a site different from the typical CBD, on the side of Gal-1 opposite to where typical β-galactosides bind [49]. Studies on Davanat derivatives are currently in a clinical trial (*α*-galactomannan—GM-CT-01) to evaluate its activities in solid tumors [39]. There are also other galectin inhibitor molecules that have biological effects that are now on clinical trials, as shown in the Appendix A (where we highlight some molecules, which galectin is an inhibitor, its biological effect and Clinical Trials), with indications for various diseases and cancer [50].

The literature indicates that different polysaccharides from *Pleurotus eryngii* have bioactivities; interestingly, MG-*Pe* has been uniquely linked to the treatment of melanoma. Of note, most of these polysaccharides contain β-1,3,6-d-linked galactans and few, such as MG-*Pe*, are α-1,6-d-Galp [51].

Miller and colleagues showed that Gal-3 also binds to galactomannans (GMs), as well as to α- and β-mannans. Nevertheless, they reported that Gal-3 binding affinity to GMs is correlated with the Man/Gal ratio, thus providing insight into structure–activity relationships and suggesting that α-(1→6)-linked Gal residues in these GMs are crucial for optimal binding, which apparently occurs at both the F-face and S-face of the lectin. Gal-3 also interacts in an unknown fashion with β-1,2-linked oligomannosides on the surface of *Candida albicans*, including galactomannans (GMs) with α-1,6-Gal branches [52].

It is possible that Gal-3 is interacting with mannogalactan in an unusual fashion, e.g., perhaps binding at different sites outside of the CBD, and this binding may interfere with the pro-tumoral action of Gal-3. In order to evaluate the involvement of the interaction between Gal-3 and MG-*Pe* in its anti-melanoma activities, we tested whether MG-*Pe* would interfere in tumor growth and tumor response to dacarbazine in tumor-bearing Gal-3 *knockout* mice.

### 2.4. The Antitumoral Effects of MG-Pe Were Mediated by Inhibition of Gal-3

In Figure 1, we observed the antitumor and adjuvant effect of MG-*Pe* in wild-type mice (WT), especially when comparing the PBS and MG-*Pe* groups (with a difference of 63%), DTIC with DTIC+MG-*Pe* (with a difference of 66%) and MG-*Pe* with DTIC (with a difference of 83%).

We followed up these experiments using knockout animals for Gal-3 (Gal-3/KO), to determine the impact of the Gal-3 deficiency on the antitumor effects of MG-*Pe*. We observed that MG-*Pe* treatment was neither effective for controlling tumor growth nor for the additive effect on DTIC treatment (Figure 4). Altogether, these data indicate for the first time that the in vivo antitumoral activities of MG-*Pe* is dependent on endogenous Gal-3. Some authors suggest that Gal-3 is among the next-generation biomarkers for detecting diseases, such as cancer [53]. The identification of novel extracellular ligands for Gal-3 is a first approach to develop novel Gal-3 inhibitors.

Although there is much to be explored, evidence indicates that Gal-3 may be involved in balancing numerous tumor cell activities during cancer development, progression, and metastasis [54]. Gal-3 interacts with and activates different molecular partners, e.g., EGFR and integrins, which are the drivers of these processes. Thereby, targeting Gal-3 functions would contribute to induce pro- or anti-tumor actions, depending on the galectin-binding partner [55]. Much remains to be studied in regard to the signaling pathways involved, e.g., examining whether the blockade of Gal-3 affects the Gal-3/EGFR/AKT/FOXO3 [56], EGFR/ERK/Runx1, BMP/smad/Id-3 and integrin/FAK/JNK axes [57].

Gal-3 is generally an intracellular protein, but under stressful conditions (tumor microenvironment) it can be secreted to an extracellular environment. After secretion, it interacts with cell surface glycoproteins, integrins, growth factor receptors, and extracellular matrix molecules. Gal-3 modifies the tumor microenvironment, favoring the migration and exit of the tumor cells from the stressed environment and the entry of endothelial cells or leukocytes, therefore, favoring cells resistance to stress, inducing migration, and favoring angiogenesis, promoting a protumoral role [58,59]. The MG-*Pe* effect is mediated by interaction with extracellular Gal3, showing that the antitumor effect was lost without its presence, reinforcing that this compound action is dependent on blocking Gal-3 extracellular protumoral actions.

### 2.5. MG-Pe Modulated Oxidative Stress in Tumor and Liver Tissue, Mediated by Gal-3

In order to evaluate the systemic effects of MG-*Pe* treatment and the involvement of Gal-3 in these effects, we further evaluated aspects of the hepatic and tumor imbalance of reactive oxygen species in the experimental conditions described. We evaluated the accumulation of oxidized lipids through the measurement of lipid hydroperoxide levels (LPO) and the consumption of reduced glutathione (GSH).

As shown in Figure 5A, tumor LPO levels increased upon treatment with DTIC, either alone or in combination with MG-*Pe*. Treatment with MG-*Pe* alone also increased LPO levels in WT animals, but not in Gal-3/KO mice. In the liver, however, MG-*Pe* led to a slighter increase in LPO levels in both WT and Gal-3/KO mice (Figure 5C). As it was observed in tumors, treatment with DTIC led to an increase in hepatic LPO, which was slightly reduced by MG-*Pe* in WT mice. The accumulation of oxidized lipids was accompanied by a reduction in GSH in WT mice, but lost in Gal-3/KO mice, both in tumor and hepatic tissues of animals treated with MG-*Pe* alone (Figure 5B,D). The consumption of GSH caused by DTIC treatment was not dependent on Gal-3. Altogether, MG-*Pe* alone led to a Gal-3-dependent imbalance in reactive oxygen species metabolism. DTIC, either alone or in combination, led to a more sustained imbalance in a Gal-3 independent manner. These results can indicate an adverse effect of DTIC, as this healthy organ can metabolize DTIC, which is a pro-drug. The literature reports that DTIC is associated with a severe and distinctive liver injury, including sinusoidal obstruction syndrome that can occur with the usual chemotherapeutic doses [60]. Although it is already known that chemotherapeutic agents, such as DTIC, lead to an imbalance in reactive oxygen species [61], this is the first demonstration that MG-*Pe* induces such an imbalance, which is associated with its antitumor effect that is lost in the absence of Gal-3.

In the normal cells, GSH has a physiological role, acting as a cellular detoxification system [62,63], but in tumor cells lower GSH levels are related with high LPO levels, generating a low antioxidant defense, and thus damage to the cellular membrane, generating lipid peroxidation, and, consequently, an antitumor action [64].

In this study, we demonstrated that MG-*Pe* modulates oxidative stress, altering GSH and LPO levels, an effect dependent on Gal-3. Changing oxidative stress via Gal-3 favors several signaling pathways. GSH analogues have been used to sensitize tumors to the cytotoxic effects of antineoplastic agents by depleting the related GSH-cytoprotective effects [65]. The increase in GSH levels is associated with cell proliferation, an essential step for the progression of the cycle of normal and malignant cells [66]. In fact, in malignant tumors, as compared with normal tissues, high levels of GSH are observed, which makes neoplastic tissues more resistant to chemotherapy [67,68], leading to cancer patients’ death [69]. Tumor cells have increased oxidative stress (more GSH) as a cellular protection [70], so in tumor tissues reducing these antioxidants (less GSH) it is interesting and leads to the death of tumor cells (high LPO).

Adaptation to oxidative stress and drug resistance is a common process in cancer. Tumor cells increase ROS due to altered metabolic demand, thus favoring a redox imbalance, altering cell signaling, and thus activating the cell’s survival mechanisms, as well as proliferation and metastasis [65]. Accordingly, in this research there was a depletion of GSH levels and an increase in tumor LPO, which could be directly related to the decrease in metastasis.

Gal-3 is a regulator of mitochondrial homeostasis, altering the mitochondrial membrane potential and the formation of ROS [71]. Studies show that Gal-3 has an apoptotic regulator [71], promotes tumor metabolic reprogramming by adapting to microenvironmental stress caused by oxygen and nutrient deprivation [72], and it is suggested that it protects cells from oxidative stress-induced cell death [73,74]. Increase Gal-3 expression can changes cell production of ROS [75] and subsequent GSH thus, in our work, Gal-3 binder maybe can consequently reduce GST and decreases antioxidant defenses. However, GST activity was not herein performed.

Although there is much to be explored, the literature shows that Gal-3 is involved in balancing numerous tumor cell activities during the development, tumor progression, and cancer metastasis. Gal-3 functions are, however, clearly dependent on its partners (surface receptors and other ligands). Thus, targeting this set would contribute to induce pro- or anti-tumor actions, depending on the galectin-binding partner [55]. In this study, we showed that the polysaccharide MG-*Pe*, previously shown to be an effective antitumor agent [15], is a direct ligand of Gal-3 resulting in modulation of oxidative stress and consequent reduction in experimental and spontaneous metastasis.

The effect of MG-*Pe* on the redox regulation is probably one of its antitumor actions but not the only one. The redox system in the tumor works different to normal cells, since neoplastic cells generate higher amounts of reactive oxygen species, often leading to deregulation of the antioxidant defense system [70]. This feature can be explored therapeutically. Herein, the increase in lipid peroxidation products (LPO) combined with the decrease in the amount of antioxidant molecules (GSH) can indicate that MG-*Pe*-induced oxidative stress in the tumor microenvironment, leading to tumor cells death. Therefore, the polysaccharide effects were more evident in wild type mice, suggesting that MG-*Pe* led to a Gal-3-dependent imbalance in redox system. We observed an increase in LPO also in the hepatic tissues of the MG-*Pe* treated groups, which denotes a mechanism not exclusive of the tumor tissue.

This mechanism is not sufficient to explain the whole phenomenon described herein and deserves further investigation, including evaluations of other oxidative stress parameters, mitochondrial activity, and cell death pathways related with oxidative imbalance. In conclusion, we characterized MG-*Pe* [15] as a novel ligand to Gal-3, that exerts antitumor activities and prevents melanoma growth and metastatic spread. MG-*Pe* also has an additive effect to DTIC on melanoma growth. The antitumor activities depend on the presence of Gal-3, and they may be related with the imbalance of oxidative species caused by MG-*Pe* in the Gal-3 expressing systems. Finally, our results show that the Gal-3 ligand MG-*Pe* modulated oxidative stress, reduced metastasis and increased survival time, generating an effective adjuvant effect in therapy alone or in combination with DTIC. Ultimately, we propose that MG-*Pe* can then be a promising strategy as an adjuvant therapeutic agent for the treatment of cancer in a combinatory therapy (synergistic effect) or alone.

## 3. Materials and Methods

A workflow of the techniques is presented in Appendix A, where we demonstrate the in vivo assays with solid tumor or lung mass; and also in vitro, with the different techniques used and described below.

### 3.1. Partially Methylated Mannogalactan (MG-Pe)

The polysaccharide used in this work is a partially methylated mannogalactan (MG-*Pe*); this heteropolysaccharide was previously isolated, as described by Biscaia et al. [15]. Briefly, MG-*Pe* was isolated by cold aqueous extraction from edible mushroom *Pleurotus eryngii* (“King Oyster”) basidiocarps, from which was obtained a mannogalactan having a main chain of (1→6)-linked α-d-galactopyranosyl and 3-*O*-methyl-α-d-galactopyranosyl residues, both partially substituted at OH-2 by β-d-Man*p* (MG-*Pe*) single-unit, a molecular weight 20.9 × 10^3^ g mol^−1^.

### 3.2. Cell Culture

B16-F10 murine melanoma cells (ATCC) were maintained with RPMI Medium 1640 (CAT 31800-022 GIBCO), supplemented with 10% (*v*/*v*) fetal bovine serum (FBS), 4.76 g/L Hepes, 2 g/L sodium bicarbonate at 37 °C in 5% CO_2_ in a humidified atmosphere. No antibiotics were used in the cell culture used for animal inoculation.

### 3.3. In Vivo Experiments and Treatment Groups

SPF (specific pathogen free) C57BL/6 strain mice, male, between 8–12 weeks old were maintained and treated, following ethical principles established by the Ethics Committee on Animal Experimentation of Faculdade de Medicina da Universidade de São Paulo (FMUSP) certificate #043/17.

#### 3.3.1. Primary Tumor Model

C57BL/6 mice (7 weeks old) were weighed and then immobilized for B16-F10 murine melanoma cells inoculation (2.5 × 10^5^ cells in 100 µL of phosphate buffered saline (PBS)) subcutaneously on the dorsal flank. After 5 days, all treatments were started via intraperitoneal injection for 10 consecutive days. The experimental groups used were PBS buffer (control group); Dacarbazine (DTIC) (40 or 80 mg/kg) (SIGMA D2390); MG*-Pe* (50 mg/kg); and the combination of DTIC+MG-*Pe* (80 mg/kg + 50 mg/kg). Animals were analyzed daily by measuring tumor diameters with a digital caliper (FORD). The volume was calculated using the formula: tumor volume (cm^3^) = (d). (D). (D). (0.52). Being: d = smaller diameter; D = larger diameter; 0.52 = ellipse volume calculation factor [76]. At the end of treatment, animals were anesthetized with ketamine (100 mg/kg) and xylazine (10 mg/kg) and euthanized.

#### 3.3.2. Survival Assay

Similar to Section 3.3.1., 5 days after cells inoculation, animals were treated daily for 10 days. Survival after that was recorded in days. If the tumor ulcerated or reached a diameter equal to or greater than 1.5 cm^3^ or the animal presented difficulties in locomotion and feeding, they were euthanized (same as in Section 3.3.1).

#### 3.3.3. Metastases Assays—Experimental and Spontaneous

For the experimental lung metastasis model, male C57BL/6 mice, 8 weeks old, were inoculated with 2.5 × 10^5^ B16-F10 murine melanoma cells, I.V. (intravenously) in the lateral tail vein. After 10 days of cell inoculation, animals were treated daily for 10 consecutive days (except DTIC, injected every 3 days, days 5, 8 and 11). The experimental groups (*n* = 5) were PBS buffer; DTIC (40 or 80 mg/kg); MG-*Pe* (50 mg/kg); and the combination of DTIC+MG-*Pe* (40 or 80 mg/kg + 50 mg/kg). Then, they were euthanized (same as in Section 3.3.1) on the 21st experimental day and the lungs were collected, *n* = 5 in each group.

For the spontaneous metastasis model, animals were inoculated with cells subcutaneously for the development of solid tumor. After 11 days (tumor volume 0.5–0.8 cm^3^), the animals were submitted to tumor removal surgery, under anesthesia with ketamine (100 mg/kg) and xylazine (100 mg/kg), added of the analgesic tramadol (20 mg/kg) (TEUTO L9068029), intraperitoneally. Under the anesthetic plan, the hair in the tumor area was removed; from the tumor area, the area was sterilized with 70% alcohol and the tumor was excised. Then, wound was sutured with the wedge closure suture, using Paralon 4-0 suture thread (Paramed sutures), with an average of 3 knots per animal, and if necessary, tissue glue (3M Vet Bond Tissue Adhesive No. 146958) was applied. During recovery, dressings were applied to the animals using Pasta D’água (ADV) and antiseptic (Chlorhexidine Gluconate 0.5% VIC pharma). After 72 h of surgery, treatments were started and lasted for 11 consecutive days. On the 32nd experimental day, the animals were reevaluated and then subjected to a thorough observation of metastasis in the axillary lymph node chain; the presence of metastatic *foci* spread throughout the animals’ bodies; the appearance of secondary tumors; the recurrence of the primary tumor; metastases in the heart; metastases in the lungs and bleeding in the spleen. Thus, the quantification of each item was performed as the percentage of appearance.

### 3.4. Galectin Binding Assay

#### 3.4.1. Expression and Purification of Recombinant Human Gal-3

Human Gal-3 was produced in *Escherichia coli* BL21/DE3 transformed with pET11a plasmid, encoding the human Gal-3 gene, LGALS3. *E. coli* cells were grown in Luria-Bertani (LB) medium supplemented with 100 µg/mL of ampicillin in an inoculum overnight at 37 °C, 180 rpm. Then, this suspension was transferred into 1 L of LB medium supplemented with 100 µg/mL of ampicillin and grown at 37 °C and 180 rpm until the optical density (OD at 600 nm) reached 0.45. Protein expression was achieved by the addition of IPTG 500 µM followed by culturing for 4 h at 37 °C, 180 rpm. Finally, the suspension was centrifuged at 10,000× *g* for 10 min at 4 °C. The obtained cell pellets were lysed (Lysis buffer: PBS, lysozyme, DNase, RNase and protease inhibitor cocktail) for 1 h at 4 °C, sonicated and then centrifuged at 10,000× *g* for 10 min at 4 °C. The soluble portion (supernatant) was collected and loaded into a lactosyl-Sepharose resin (Sigma) followed by a wash with PBS supplemented with 14 mM of 2-mercaptoethanol (PBS-2-ME). Gal-3 was eluted with PBS-2-ME supplemented by 100 mM lactose and quantified in NanoDrop 2000 device using the NanoDrop 2000 software, version 1.4.1, (Thermo Fisher Scientific, Wilmington, NC, USA). Lactose and 2-ME were removed from protein preparations using a PD-10 gel-filtration column, and the Gal-3 solution in PBS was kept at 4 °C until use.

#### 3.4.2. QCM (Quartz Crystal Microbalance)

MG-*Pe* and Gal-3 binding was analyzed in a QCM-D E4 instrument (Biolin Scientific AB, Gothenburg, Sweden). On the first day, the gold quartz crystals sensors (Qsensor QSX301 Gold -QSense) were exposed in UV for 10 min; after it were immersed in basic piranha solution 5:1:1 H_2_O/NH_4_OH/H_2_O_2_, for 5 min; washed twice with ultrapure water; dried; UV for 10 min; and immersed overnight (O/N) in a *β*-mercaptoethanol 0.05 M (Sigma M3148) solution. The next day, they were washed in absolute ethanol and placed in QCM-D flow chamber apparatus connected to a syringe pump with 100 µL.min^−1^ flow rate (KD Scientific). A water flux (room temperature—R.T.) was used for initial hydration, until stabilization. Next the reaction was done with TEMPO solution (SIGMA CAT 214000) and EDC/NHS [EDC ((N-(3-Dimethylaminopropyl)-N′-ethylcarbodiimide hydrochloride))—Sigma CAT E7750—100 mM]; [NHS (N-Hydroxysuccinimide) Sigma CAT 130672—400 mM] in acetate buffer (Alphatec 6131-90-4) pH 4.5–5.0. QCM sensors were removed and 100 µL of 100 µg/mL Gal-3 (item 3.4.1.) was added in each sensor, and stayed in a humid chamber O/N. It was washed twice in PBS (saline buffer) and placed again in the QCM flow chamber on the third day. The solution with MG-*Pe* was injected, the frequency and dissipation were observed, and the data (ng/cm^2^) were evaluated using the QSense DFind Software. The results represent three independent experiments and present the average of three harmonics of each group.

#### 3.4.3. AFM (Atomic Force Microscopy)

After QCM-D assay, sensors were dried with a gentle nitrogen flow. The AFM images were obtained in air in the intermittent contact mode with an Agilent 5500 microscope (Agilent Technologies, Santa Clara, CA, USA) using silicon tips NSC15 (Mikromasch USA, San Jose, CA, USA), with a resonance frequency of ~325 kHz and a nominal spring constant of 40 N/m. The scanned areas were 2.0 µm × 2.0 µm. Images were acquired using Pico Image software v. 1.14 (Agilent Technologies, Santa Clara, CA, USA) and were processed with the Gwyddion software (Czech Metrology Institute). The Student’s *t*-test was used and a significant difference of *p ≤* 0.01 was assumed.

#### 3.4.4. Contact Angle (CA) Tensiometer

After acquiring AFM images, the contact angle (CA) between water and the modified QCM-D crystal surface was measured with a tensiometry instrument (DataPhysics OCA 15+ tensiometer, Filderstadt, Germany), using a 500 µL Hamilton syringe and needle with outside diameter of 1.65 mm and 38 mm length. The pendant mode method was used with ultrapure water droplets of 5 µL.

All measurements were conducted in an environment of 22.4 °C and 63.9% relative humidity. The CA calculations were performed with SCA 20 DataPhysics software (Filderstadt, Germany) and the average was used.

### 3.5. Effect of MG-Pe via Gal-3

To investigate the influence of Gal-3 on the MG-*Pe* antitumoral effect, Gal-3 knockout on C57BL/6 mice (Gal-3/KO [77]) and wild C57BL/6 mice (wild type—WT) were used as a control. Four experimental groups were used: PBS (vehicle); DTIC (80 mg/kg); MG-*Pe* (50 mg/kg) and the combination of DTIC+MG-*Pe* (80 + 50 mg/kg).

At the time of inoculation, as described in item 3.3.1., an aliquot of B16-F10 cells was separated and then the Western Blotting assay for Gal-3 detection in the cells was performed, and SK-MEL-28 cells were used as a positive control. So, we could verify that there was no Gal-3 in the cells at the time of inoculation, as shown in the figure by the absence of the band in B16-F10 cells (Appendix A).

### 3.6. Oxidative Stress Assay

After the experiments described in Section 3.5, the tumor and liver samples were collected separately and homogenized in a 1:10 dilution in potassium phosphate buffer (0.1 M, pH 6.5). Subsequently, 100 µL were separated, suspended in 80 µL of trichloroacetic acid (12.5%), and vortexed and centrifuged at 6000× *g* for 15 min at 4 °C for analysis of reduced glutathione (GSH) levels, according to Sedlak and Lindsay [78]. The remaining homogenate was centrifuged at 9700× *g* for 20 min at 4 °C to determine lipoperoxidation (LPO), as described by Jiang et al. [79].

### 3.7. Statistical Analyses

Statistical analyses were performed using software Graph Pad Prism 6.0 (Graph Pad Software^®^, Inc., San Diego, CA, USA). Data analysis was performed and the tests was showed in each subtitle, where * *p* < 0.05 was considered statistically significant.

## 4. Conclusions

Importantly in our study, we found that the MG-*Pe*, previously shown to be an effective antimelanoma agent [15], is a novel ligand to Gal-3 that exhibits adjuvant effects to DTIC on melanoma growth and survival animal rates. Moreover, MG-*Pe*’s antitumor activities are linked to the inhibition of Gal-3, resulting in the modulation of oxidative stress and consequent reduction in experimental and spontaneous metastasis, and, for the first time, we have shown that Galectin-3 mediates the antitumor action of this polysaccharide.

## 5. Patents

This work has a patent application in Brazil, entitled “Antitumor activity of partially methylated mannogalactan from *Pleurotus eryngii* mushroom”, referring to process number BR 102018 005595-0 A2 [80].

## Figures and Tables

**Figure 1 ijms-23-07635-f001:**
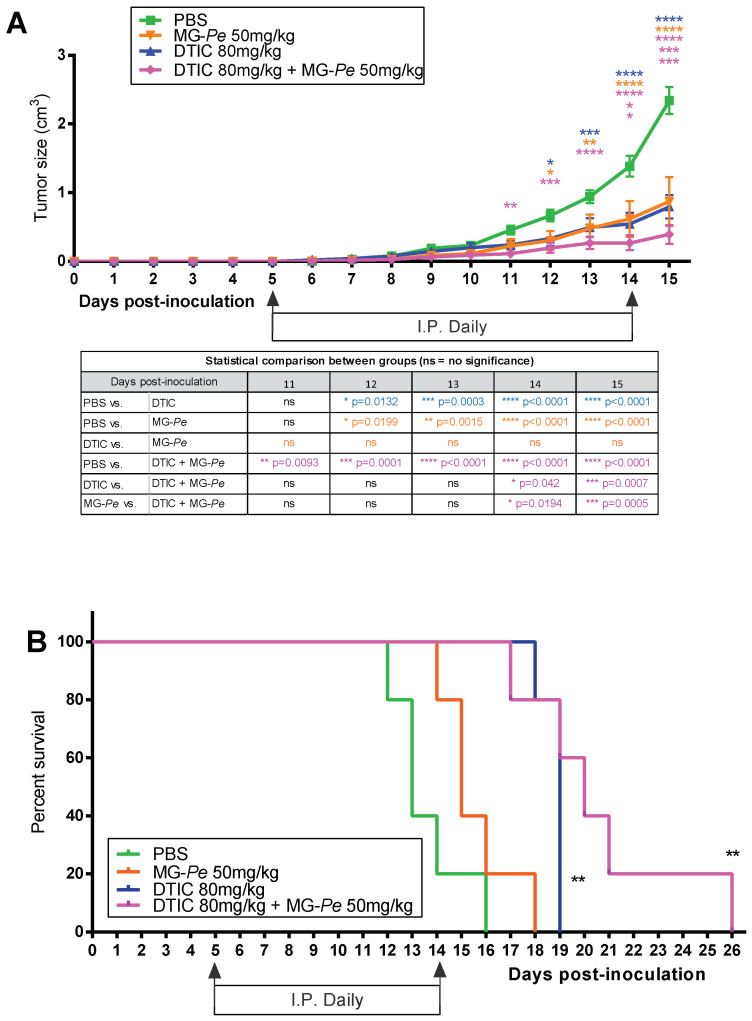
Partially methylated mannogalactan extracted from mushroom *Pleurotus eryngii* (MG-*Pe*) has an adjuvant effect with Dacarbazine (DTIC) and increases animal survival. Tumor development (**A**) and survival experiment (**B**). C57BL/6 tumor bearing mice (B16-F10 cells) were I.P. daily treated for 10 days with PBS (saline control), DTIC (80 mg/kg), MG-*Pe* (50 mg/kg) alone, or in combination DTIC+MG-*Pe* (80 + 50 mg/kg). (**A**) *n* = 5 animals/group (mean/SEM), analyzed by two-way ANOVA test with Tukey’s post-test with multiple comparisons; (**B**) *n*=5 animals in each group, analyzed by survival curve test, ** *p* ≤ 0.01.

**Figure 2 ijms-23-07635-f002:**
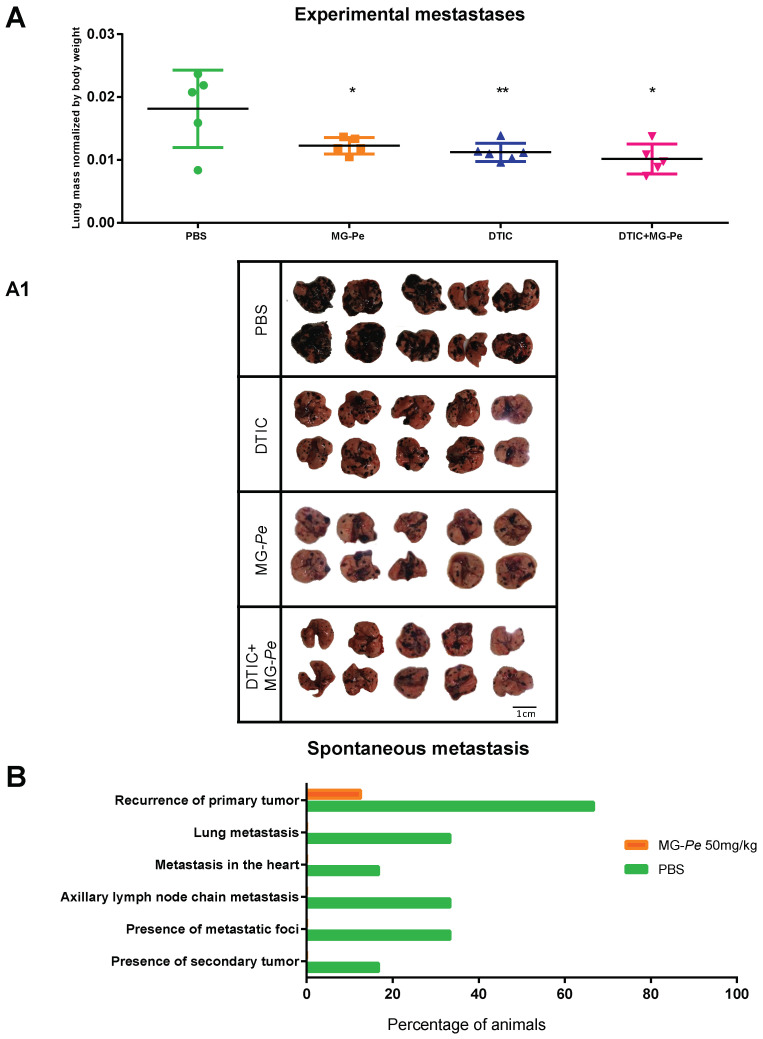
Analysis of experimental and spontaneous metastases of B16-F10 cells in C57BL/6 mice. (**A**) Experimental metastasis model: cells were intravenously inoculated via lateral caudal vein and treated with: PBS (*n* = 5), DTIC (40 mg/kg) (*n* = 6), MG-*Pe* (50 mg/kg) (*n* = 6), or DTIC+MG-*Pe* combination (40 + 50 mg/kg) (*n* = 5). The data (mean/SD) were submitted to *t*-test. * *p* ≤ 0.05 and ** *p* ≤ 0.01. (**A1**) Lung images: dorsal (up) and ventral (down). (**B**) Spontaneous metastasis model: cells were subcutaneously inoculated; animals were treated for 10 days with PBS (*n* = 6) or MG-*Pe* (50 mg/kg) (*n* = 8); then tumor removal by surgery (21st day); waited 1 month (without treatment), and finally tumor recurrence was assessed.

**Figure 3 ijms-23-07635-f003:**
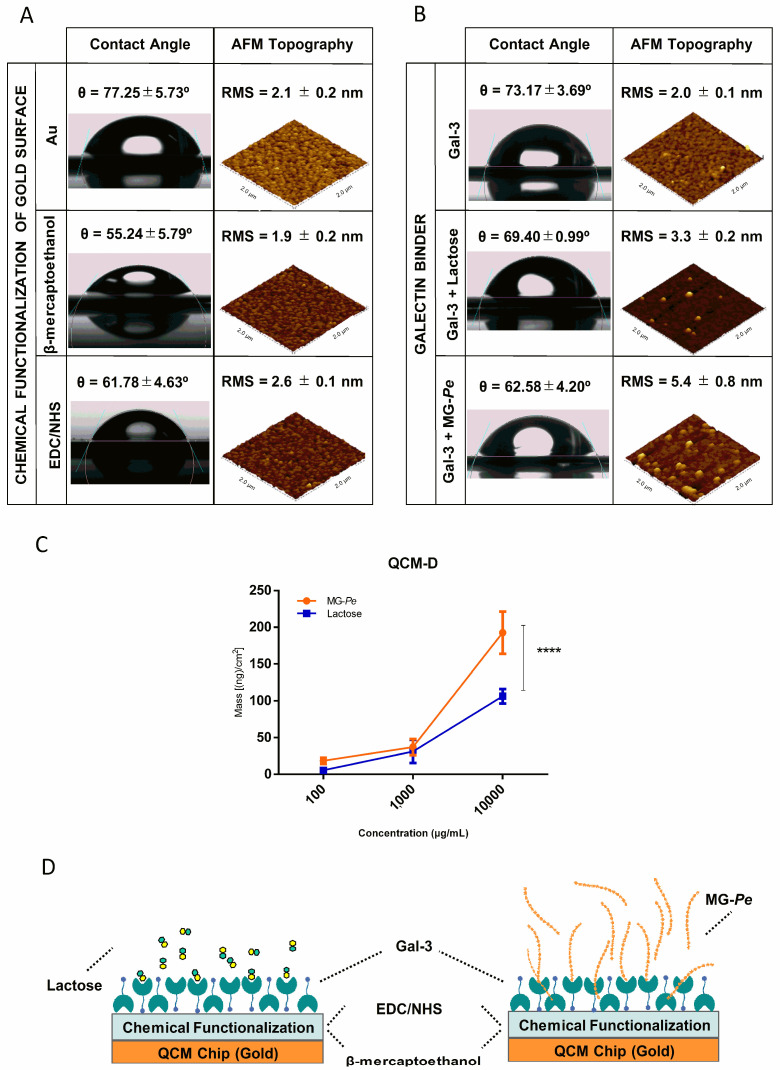
MG-*Pe* is a Gal-3 Binder. (**A**) Chemical functionalization of gold surface; (**B**) Gal-3 and its ligands—contact angle and AFM (atomic force microscopy) (2.0 × 2.0 µm scale); (**C**) QCM-D (quartz crystal microbalance with dissipation monitoring). Analyzed by two-way ANOVA test with Sidak with multiple comparisons; **** *p* ≤ 0.0001; and (**D**) schematic representation of a QCM-D crystal surface.

**Figure 4 ijms-23-07635-f004:**
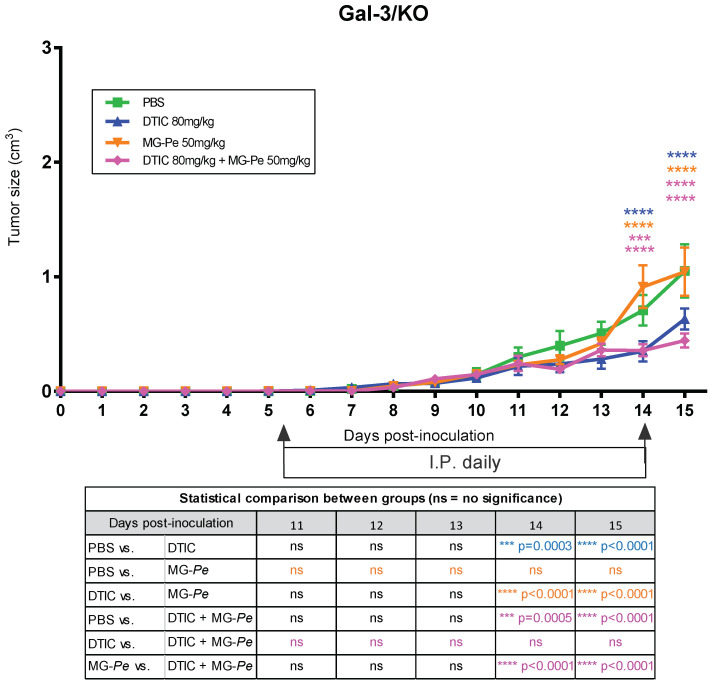
Tumor evolution with the influence of Gal-3. To investigate Gal-3 influence on MG-*Pe* antitumoral effect C57BL/6 Gal-3 *knockout* mice (Gal-3/KO) were used. Mice were inoculated with B16-F10 cells on the right dorsal flank, and subsequent treatment with PBS (control), dacarbazine (DTIC) (80 mg/kg), MG-*Pe* (50 mg/kg), or the combination of DTIC+MG-*Pe* (80 + 50 mg/kg). The data were submitted to a two-way ANOVA test with Tukey’s post-test with multiple comparisons (mean/SEM) (*n* = 5 in each group).

**Figure 5 ijms-23-07635-f005:**
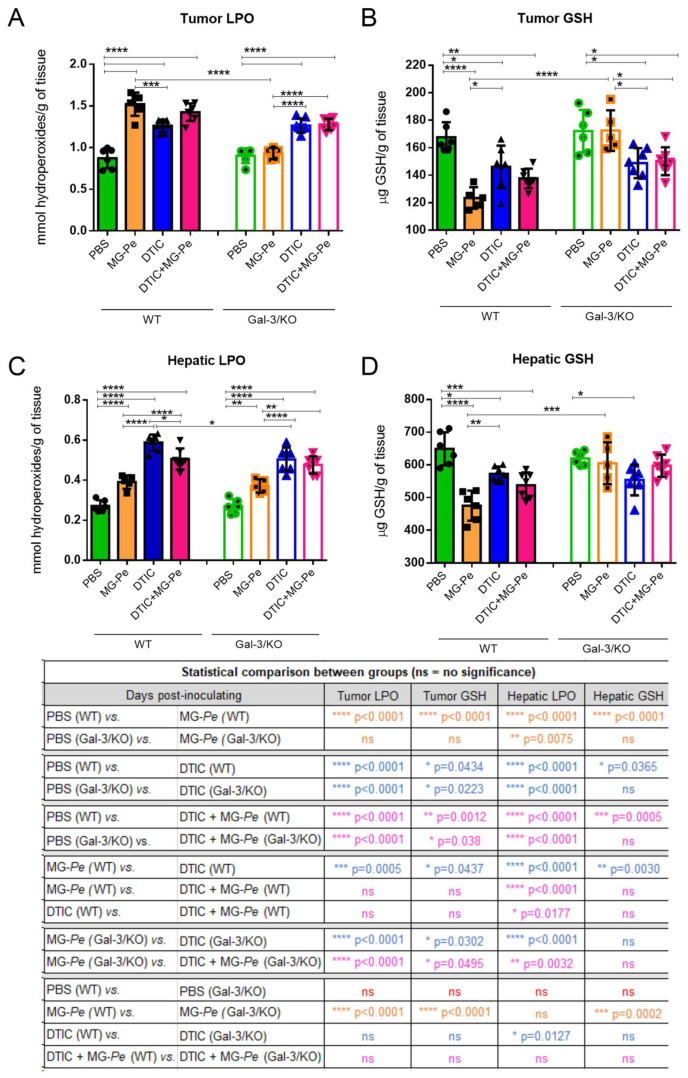
Tumor Effects of MG-*Pe* in oxidative stress. Tumor (**A**,**B**) and hepatic (**C**,**D**) levels of reduced glutathione and lipoperoxidation from wild mice—WT (C57BL/6) or Galectin-3 knockout mice (C57BL/6 Gal-3/KO). Animals were treated with PBS (control, *n* = 6), partially methylated mannogalactan (MG-*Pe* 50 mg/kg, *n* = 6), Dacarbazine (DTIC 40 mg/kg, *n* = 7) or with Dacarbazine+MG-*Pe* (DTIC+MG-*Pe*, *n* = 7). Statistical analysis was performed by one-way ANOVA, followed by Dunnet’s post-test. Values are presented as mean ± SEM; * *p* ≤ 0.05, ** *p* ≤ 0.01, *** *p* ≤ 0.001, **** *p* ≤ 0.0001.

## Data Availability

Supporting Data are available at https://doi.org/10.1016/j.carbpol.2017.08.117 (accessed on 4 July 2022).

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
