# Peer review of "MG-Pe: A Novel Galectin-3 Ligand with Antimelanoma Properties and Adjuvant Effects to Dacarbazine"

_ijms, 2022, doi:10.3390/ijms23147635_

Round 1

Reviewer 1 Report

The authors have previously isolated and characterized a partially methylated mannogalactan (MG-Pe) and showed that it had antimelanoma activities. In this paper they showed that this polysaccharide binds galectin-3, and its antimelanoma activities rely on host Gal3 expression.

Galectins are recognized more and more as regulators of cancer formation, progression, and metastasis by regulating cell-cell interaction, cell death and immunity. The findings of MG-Pe as a gal3 inhibitor that suppress melanoma growth and metastasis are therefore interesting. However, there are some major concerns that need to be addressed.

Major points:

1.     Both Fig. 3b and c need sucrose as a negative control to determine background signal.

2.     Galectins share common binding affinity for b-galactosides. Therefore, other galectins also need to be tested for MG-Pe binding to assess specificity.

3.     General ack of mechanism. The authors show in Fig 5 that MG-Pe treatment increased oxidative stress in tumor and liver, but it is not known how this is related to its tumor suppression activity.

4.     Gal3 is mostly an intracellular protein. How extracellular MG-Pe inhibit intracellular Gal3 need to be discussed. Can secreted or surface expressed gal3 be detected in the TME or under experimental conditions? Source of Gal3 in the TME also need to identify, as B16 melanoma cells do not express this protein (suppl material 1).

Minor points:

1.     Source of Gal3 KO mice was not described.

2.     Miss out key relevant papers in the literature.

Reviewer 2 Report

The manuscript entitled; MG-Pe: a novel Galectin-3 ligand with anti-melanoma properties and adjuvant effects to dacarbazine, was reviewed. It is a very well-designed study with promising results in the field that developing new therapeutic is extremely crucial. Gal-3 has a lection has several roles in physiological settings and pathological conditions and the result of this manuscript push the field forward. Here I have only few comments to increase the impact of the very interesting findings.

1-    It would be very useful for readers to provide a table and summarize different Gal-3 inhibitors that have been developed so far for melanoma and other cancers. It would be great if you can provide their pharmacology class and mention that how advanced they are in clinician trials. I understand that this manuscript is research article but providing this information help the readers to have more comprehensive viewpoint.

2-    It would be very helpful if author could provide a schematic figure to show the workflow of their methods because of the diversity of the methods that they used

3-    Gal-3 has a very important role in CNS especially the pathophysiology of neurodegenerative and neuroinflammatory diseases. Several interesting research and review articles have addressed the multimodal effects of Gal-3 (Med Res Review 2021, Mol Neurobiology 2019 and Drug Discovery Today 2018). These findings also imply the importance of Gal-3 in the future drug discovery of brain malignancies. I think explaining these findings in your discussion increases the depth of manuscript and gives the expert people more insight about different aspects of Gal-3.
